# Influence of Microbiota-Related Metabolites Associated with Inflammation and Sepsis on the Peroxidase Activity of Cyclooxygenase in Healthy Human Monocytes and Acute Monocytic Leukemia Cells

**DOI:** 10.3390/ijms242216244

**Published:** 2023-11-13

**Authors:** Natalia Beloborodova, Roman Fadeev, Nadezhda Fedotcheva

**Affiliations:** 1Federal Research and Clinical Center of Intensive Care Medicine and Rehabilitology, 25-2 Petrovka St., 107031 Moscow, Russia; nvbeloborodova@yandex.ru; 2Institute of Theoretical and Experimental Biophysics, Russian Academy of Sciences, 3, Institutskaya St., 142290 Pushchino, Russia; fadeevrs@gmail.com

**Keywords:** cyclooxygenase, COX activity, monocytes, inflammation, microbial metabolites, 4-hydroxyphenyllactic acid, itaconic acid, tert-butyl hydroperoxide, thiol reagents, TMPD

## Abstract

The human microbiota produces metabolites that can enter the bloodstream and exert systemic effects on various functions in both healthy and pathological states. We have studied the participation of microbiota-related metabolites in bacterial infection by examining their influence on the activity of cyclooxygenase (COX) as a key enzyme of inflammation. The influence of aromatic microbial metabolites, derivatives of phenylalanine (phenylpropionic acid, PPA), tyrosine (4-hydroxyphenyllactic acid, HPLA), and tryptophan (indolacetic acids, IAA), the concentrations of which in the blood change notably during sepsis, was evaluated. Also, the effect of itaconic acid (ITA) was studied, which is formed in macrophages under the action of bacterial lipopolysaccharides (LPS) and appears in the blood in the early stages of infection. Metabiotic acetyl phosphate (AcP) as a strong acetylating agent was also tested. The activity of COX was measured via the TMPD oxidation colorimetric assay using the commercial pure enzyme, cultured healthy monocytes, and the human acute monocytic leukemia cell line THP-1. All metabolites in the concentration range of 100–500 μM lowered the activity of COX. The most pronounced inhibition was observed on the commercial pure enzyme, reaching up to 40% in the presence of AcP and 20–30% in the presence of the other metabolites. On cell lysates, the effect of metabolites was preserved, although it significantly decreased, probably due to their interaction with other targets subject to redox-dependent and acetylation processes. The possible contribution of the redox-dependent action of microbial metabolites was confirmed by assessing the activity of the enzyme in the presence of thiol reagents and in model conditions, when the COX-formed peroxy intermediate was replaced with *tert*-butyl hydroperoxide (TBH). The data show the involvement of the microbial metabolites in the regulation of COX activity, probably due to their influence on the peroxidase activity of the enzyme.

## 1. Introduction

Currently, much attention is being given to the components of microbial cells, including lipopolysaccharides (LPS), that are detected by the innate host immune system throughout the body. LPS are a structural component of the cell wall of Gram-negative bacteria, but a number of other compounds, small molecules originating from microbes (SMOMs), are the products of the metabolism of living bacteria, and above all, the microbiota [1]. The human microbiota produces metabolites that may enter the bloodstream and exert systemic effects on various functions of the human organism in both healthy and pathological states. Microbial metabolites have been implicated in the regulation of the immune system, the central nervous system, metabolism, and epigenetic control [2,3,4]. They regulate the immune system via several ways, including receptor and metabolic signaling pathways [5]. It is believed that pathological conditions cause microbiota dysbiosis and disturb the production of microbial metabolites, which leads to the dysregulation of the immune system and metabolism [5,6]. Additionally, these relationships are bidirectional, since gut microbiota dysbiosis may provoke a high risk of the onset of chronic diseases.

Under conditions of dysbiosis, bacteria themselves may provide a proinflammatory effect through the influence of LPS, which are a part of the outer membrane of Gram-negative bacteria. In the case of an increased translocation of intestinal bacteria in the bloodstream, LPS trigger systemic inflammation and stimulate the synthesis of pro-inflammatory cytokines, namely interleukins IL-6, IL-1, and IL-27, as well as the tumor necrosis factor TNF-α [6]. In addition, LPS induce the biosynthesis of itaconic acid (ITA), which is also an important link in processes associated with macrophage-mediated inflammation [7]. It has recently been shown that itaconic acid lowers macrophage inflammation, diminishing the production of inflammatory agents, TNF-α, and the cyclooxygenase COX-2 [8], and ameliorates autoimmunity by regulating the imbalance of T cells via metabolic and epigenetic reprogramming [9].

Cyclooxygenase is one of the key participants in the inflammatory process. It enables both the inclusion of molecular oxygen into arachidonic acid to form the cyclic peroxy endoperoxide PGG2 and the subsequent reduction of PGG2 into the hydroxy endoperoxide prostaglandin H2 (PGH2), a precursor of all prostaglandins and thromboxanes. The formation of hydroxy endoperoxide occurs with the involvement of glutathione as a reducing cofactor. Prostaglandins play a key role in the generation of the acute inflammatory response; in addition, they promote chronic inflammation via participation in processes such as the amplification of cytokine signaling, the induction of the release of cytokines, and the infiltration of inflammatory cells at the inflamed site [10,11].

Some studies showed the involvement of microbial metabolites in the regulation of COX activity. Short-chain fatty acids were found to influence the immune response by stimulating the biosynthesis of prostaglandins in human monocytes, and their effect could increase in the presence of LPS. In addition, they inhibited the LPS-induced production of interleukin-10 (IL-10), TNF-α, and interferon-gamma (IFN-y), which indicates their ability to produce both pro- and anti-inflammatory effects [12]. It is assumed that the effect of short-chain fatty acids is related to their influence on the acetylation/deacetylation of transcription factors involved in the production of inflammatory cytokines [13,14]. It was found that microbial metabolites, in particular butyrate, propionate, D-lactate, and some phenolic acids, participate in the regulation of acetylation, inhibiting deacetylation reactions [15]. In addition, the intestinal phenolic acids identified in fecal water 3-phenylpropionic acid, 3-hydroxyphenylacetic acid, and 3-(4-hydroxyphenyl)-propionic acid decreased the COX-2 protein level in the human colon adenocarcinoma cell line HT-29 by up to 63% [16]. The participation of phenolic acids in the modulation of the immune response is also evidenced by our data showing their effect on neutrophils, the main producer of ROS in circulation. It was found that, among the phenolic acids tested, phenyllactate and 4-hydroxyphenyllactate most significantly decreased ROS production in PMA-activated rat neutrophils [17]. 

The basic conception of our investigations on this subject is that the metabolites of the microbiota of a healthy person have been implicated in the regulation of inflammation due to their ability to control the immune process. In sepsis, under conditions of severe dysbiosis, some microbial metabolites enter the bloodstream abundantly, which can influence the development and progression of systemic inflammation. In the present study, we evaluated the influence of some selected compounds from various groups of microbiota-related metabolites on the activity of COX. Among these are the microbial derivatives of phenylalanine and tryptophan: phenylpropionic (PPA), 4-hydroxyphenyllactic (HPLA), and indolacetic acids (IAA), the concentrations of which change in the blood during sepsis [17]; itaconic acid (ITA), which is formed in macrophages under the action of bacterial LPS and is detected in the blood in the early stages of infection [18]; and metabiotic acetyl phosphate (AcP) as a strong acetylating agent. Their influence on COX activity was tested using the commercial pure enzyme, cultured healthy monocytes, and the human acute monocytic leukemia cell line THP1. 

## 2. Results

Firstly, we tested the influence of microbial metabolites on the activity of the pure enzyme to identify the most active among them that are involved in various ways in the inflammatory process. Figure 1 shows the effect of microbial metabolites on COX activity, determined by the oxidation of TMPD, which acts as an electron donor, the intensity of color and adsorption of which increase as it oxidizes. All metabolites tested decreased enzyme activity, with the degree of inhibition ranging from 10 to 45% (Figure 1a). As shown in Figure 1b, at concentrations of 100 µM and 500 µM and after a 10 min of incubation of the enzyme with arachidonic acid and cofactors, phenylpropionic acid (PPA) decreased the optical density of TMPD by 10% and 15%, that of 4-hydroxyphenyllactic (HPLA) by 17–25%, and that of indolacetic acids (IAA) by 20–25%. The effect of itaconic acid (ItA) was close to these values (20–25% inhibition), while the influence of acetyl phosphate (AcP) was much more pronounced, reaching 45% inhibition. Figure 1c shows COX activity measured via the rate of TMPD oxidation per minute in the control and in the presence of 100 µM microbiota-associated metabolites.

In the next experiments, we evaluated the role of the redox-dependent reactions in changes in enzyme activity. For this purpose, we used the thiol antioxidant dithiothreitol (DTT) and the thiol-blocking agent N-ethylmaleimide (NEM), the redox effects of which were similar to those shown by some microbial phenolic acids on neutrophils and mitochondria [17]. As seen in Figure 1d, NEM activated and DTT inhibited the activity of the enzyme, which indicates the possible influence of the tested microbial metabolites through thiol-dependent pathways.

The influence of microbiota-associated metabolites on COX activity was studied on human healthy monocytes and the human acute monocytic leukemia cell line THP1, activated by phorbol myristate acetate (PMA) or LPS. The lysates responded to the addition of the COX substrate’s arachidonic acid with an increase in TMPD adsorption and to the addition of the COX inhibitor SC-560 with a decrease in TMPD adsorption (Figure 2a). However, these cells differed greatly in their sensitivity to TMPD-based detection. To achieve approximately equal values of TMPD adsorption, one order of magnitude more THP1 cells were required compared to healthy monocytes so that a lysate from 1 million healthy cells and a lysate from 15–20 million THP1 cells were used in each sample (Figure 2b). This ratio remained the same, although we used various methods to obtain cell lysates, including hypotonic lysis and ultra-sonication to disrupt cells.

As shown in Figure 2d, the effect of microbial metabolites on COX activity in both lysates was weaker than that on the pure enzyme; it slightly increased with an increasing concentration of them and did not exceed a 10–20% deviation from the control. Some of the metabolites reversed their effect as the concentration increased; thus, instead of inhibition, an increase in activity was observed. This refers to IAA, AcP, and ItA, which at a concentration of 500 µM caused an increase in TMPD oxidation. 

Thiol reagents had the same effect on cell lysates as that on the pure enzyme. Both compounds had a strong opposite effect on the activity of the enzyme in cell lysates; namely, NEM activated and DTT inhibited TMPD oxidation (Figure 2c). However, healthy monocyte lysates and THP1 monocyte lysates differed in sensitivity to NEM. As shown in Figure 2c, DTT almost equally inhibited TMPD oxidation in both types of lysates, while NEM activated TMPD oxidation by 15% in healthy monocytes and by 70% in THP1 monocyte lysates. It follows from these data that the redox state of thiol groups in these lysates and, accordingly, cells, varies significantly, which can affect COX activity. Since the most redox-dependent COX reaction is the glutathione- and heme-dependent peroxidase step, the influence of microbial metabolites in the following experiments was evaluated under model conditions imitating the peroxidase reaction.

We suggested that *tert*-butyl hydroperoxide (TBH) can serve as an analog of the peroxy intermediate PGG2 formed during the cyclooxygenase step and can replace PGG2 in the peroxidase step. As shown in Figure 3, TBH indeed replaced the substrate in the assay with TMPD. TBH induced TMPD oxidation, which depended on TBH concentration, required the presence of a heme, and occurred in the absence of the enzyme, i.e., in a nonenzymatic manner. The reaction was initiated by TBH addition. Thus, these model conditions reproduced the peroxidase step of the enzyme and could be applied in our experiments to evaluate the role of the redox-dependent influence of microbial metabolites on COX activity. It can be seen that none of the components by itself causes the oxidation of TMPD. The same applies to microbial metabolites, each of which was tested separately (Figure 3b). As shown in Figure 3c,d,e, microbial metabolites affected TMPD oxidation to different degrees, depending on their concentrations and the concentration of TBH. At high concentration of TBH (500 μM), all metabolites increased the rate of TMPD oxidation (Figure 3c). At a moderate concentration of TBH (250 μM), they decreased the rate of TMPD oxidation (Figure 3d). At lower concentrations of both components (TBH and a metabolite), the rate of TMPD oxidation varied in the range of 15–20% both towards activation and inhibition (Figure 3e). Furthermore, the ratio of concentrations of metabolites and hydroperoxide was of importance. Despite the variations, AcP, ItA, and HPLA can be considered the most active metabolites (Figure 3f).

## 3. Discussion

The development of a systemic inflammatory reaction in the human body can be observed in various nosological forms of disease, including local infections of the skin and soft tissues, pneumonia, osteomyelitis, and others, as well as upon tissue damage (trauma and surgery). An increase in clinical and laboratory signs of inflammation is always regarded as a progression of the pathological process, since it is associated with the development of certain complications; in the most severe cases, it is accompanied by the development of multiple organ dysfunctions and sepsis with the risk of an unfavorable outcome. Given the extensive evidence on the role of the structural components of the bacterial cell wall (LPS, etc.), as well as immunological, genetic, and many other etio-pathogenetic factors involved in triggering systemic inflammation and sepsis, today, very little is known about the role of the products of microbial metabolism in the regulation of the inflammatory process. 

Our results showed the participation of microbiota-related metabolites in the regulation of the activity of COX as a key enzyme of inflammation. Their action depends on the concentration, the initial activity of the enzyme, and the redox state of cells. As we have shown earlier, phenolic and indolic acids of microbial origin participate in the regulation of ROS production in both the circulation and tissues, and affect the activity of oxidative enzymes and membrane permeability [17,18]. Their effects strongly increase in the presence of iron ions and in acidosis inherent in infectious diseases and sepsis [19,20]. These observations also apply to itaconic acid, which affects redox-dependent enzymes and processes [20,21]. Thus, these data served as the basis for evaluating their influence on the activity of the key inflammatory enzyme COX in monocytes, the activation of which is dependent on bacterial LPS and is accompanied, among other things, by the production of itaconic acid in macrophages.

As our data show, all metabolites tested decreased the activity of the pure enzyme. Their effect diminished on lysates of both types of monocytes, healthy and THP1. Interestingly, a similar difference was observed earlier when studying the influence of some fecal microbial phenolic acids, such as 3-phenylpropionic acid, 3-hydroxyphenylacetic acid, and 3-(4-hydroxyphenyl)-propionic acid, on the activity of the pure enzyme, and the human colon adenocarcinoma cell line HT-29 [16]. Then, it was shown that these phenolic acids inhibited the biosynthesis of prostaglandin by the purified enzyme and did not affect its production in cancer cells. It was suggested that phenolic acids affect the peroxidase activity of COX. Our data obtained under model conditions are consistent with this assumption and also indicate the reasons for the observed variations.

The model system, in which the intrinsic intermediate of the cyclooxygenase step, hydroperoxy-endoperoxide PGG2, was replaced by *t*-butyl-hydroperoxide, made it possible to identify the conditions that promote either the inhibition or activation of the enzyme by these metabolites. The strong dependence of their effect on the TBH concentration suggests that the influence of metabolites is directed to the peroxidase step of the enzyme and, therefore, depends on the amount of hydroperoxide formed in the cyclooxygenase step. Also, the model conditions show a strict dependence on the presence of the heme. As known, the heme is a key participant of both steps of the cyclooxygenase reaction [22]. In the model system, the heme iron is required for an interaction with hydroperoxide by the type of Fenton reaction, while TMPD donates electrons to this system.

There are several types of reactions of phenolic compounds with iron. These are redox reactions with the participation of iron ions, where phenol-containing acids can act as reducing agents [23]. Phenolic acids containing hydroxyl groups in the phenolic ring have been shown to bind iron ions to form complexes in a 3:1 ratio, which suggests that they can act as iron chelators [24]. In addition, there is evidence indicating that the indole derivatives indole-3-propionic acid and 5-hydroxy-indole-3-acetic acid in the concentration range of 0.25–4.0 mM can act as the inhibitors of lipid peroxidation induced by iron ions [25]. Therefore, phenolic and indolic acids, being iron chelators and antioxidants, can act as the inhibitors of COX activity. Probably, this effect shows up in our experiments on the pure enzyme and at moderate concentrations of TBH in model conditions. On the contrary, the activation of the peroxidase reaction by metabolites, observed at high concentrations of TBH, may be associated with enhanced interactions of all three redox-active components, namely iron, hydroperoxide, and the metabolite. Then, the inhibition or the activation of the enzyme depends on the ratio of their concentrations in each individual case, as indicated by the variations observed at different concentrations of TBH and metabolites. 

In the context of the key role of heme iron in this reaction, it can be noted that itaconate also alters iron metabolism in macrophages by degrading Fe–S clusters in both mitochondrial and cytosol aconitase, as has recently been shown [26]. Therefore, itaconic acid can affect COX activity not only as a redox-active molecule, but also as a modulator of free iron. Moreover, itaconate could directly modify cysteine sites on functional proteins involved in inflammation [27]. 

Until recently, there was no evidence on the presence of AcP in the human blood. In bacteria, AcP is present in millimolar concentrations [28]. It was also identified in mammalian mitochondria as an intermediate of the pyruvate dehydrogenase complex [29]. Recently, an elevated level of AcP in the blood was found in pathologies associated with spinal cord injury and interpreted as an indicator of myopathy and mitochondrial breakdown [30]. In our experiments, AcP inhibited the activity of the pure enzyme, which can be explained via the acetylation of COX in a manner acetylsalicylic acid does. However, AcP affected TMPD oxidation under model conditions in the absence of the enzyme, and the direction of its effect depended on the concentration of TBH. These data show the possible involvement of AcP in iron-dependent reactions, which is consistent with our previous data on the opposite effects of AcP on membrane permeability in normal conditions and in the presence of iron ions [19]. 

These data support the assumption that bacterial-derived metabolites are involved in the control of inflammatory responses via the regulation of the cell redox balance. As supposed, a balanced microbiome provides a stable redox balance, whereas dysbiosis disturbs the redox status, thereby influencing the immune system and inflammatory responses [31]. This concept is also confirmed by our data on differences in the redox states of healthy and leukemic cells. Our experiments revealed significant differences between healthy and leukemic monocytes in their sensitivity to redox-active compounds including NEM, DTT, and TMPD. The low sensitivity of THP-1 cells to TMPD and the high sensitivity to NEM may indicate an elevated redox status of THP-1 cells. According to current views, the redox state determines the response of tumor cells, including the immune response, to various drugs and treatments. It is assumed that the level of antioxidant defense, including the protection of thiol groups, allows them to obviate the deleterious effects of inductors of oxidative stress [32]. To what extent these indicators are applicable to THP-1 cells as a whole and to COX activity in particular requires further research. However, our results obtained under model conditions show a strong dependence of the direction of the reaction on the concentration of redox active participants. As was recently shown, all isoforms of glutathione peroxidase can reduce not only fatty acid-derived hydroperoxides, but also small hydroperoxides such as TBH [33]. If the intrinsic glutathione is considered instead of the artificial electron donor TMPD, then it can be assumed that the oxidized state of cells would decrease glutathione-dependent prostaglandin synthesis, and, vice versa, the reduced state would promote it, which also needs clarification.

The results of clinical studies in different groups of patients confirm the above. It has been shown that normally, with a preserved microbiome, the intestinal contents of healthy people are dominated by the end products of microbial metabolism of aromatic amino acids (phenylpropionic acid, phenylacetic acid, and indolacetic acid) and, accordingly, they can be detected in the systemic bloodstream. They are formed from aromatic amino acids as a result of biotransformation by strictly anaerobic intestinal bacteria. Conversely, intermediate products (4-hydroxyphenyllactic acid and 4-hydroxyphenylacetic acid) are predominantly the metabolites of facultative anaerobes (e.g., Gram-negative enterobacteria (*Klebsiella pneumonia*, *Escherichia coli*, and *Staphylococci* spp.), which multiply well in the presence of oxygen and actively participate in the development of postoperative complications (pneumonia, bacteriemia, sepsis, etc.) [17]. These metabolites are absent in the intestines; they do not enter the bloodstream and have no effect on the immune system. On the contrary, in patients with documented bacterial inflammation, hydroxylated phenolic metabolites are always detected in the blood [34,35]. In the case of local forms of infection, the level of hydroxyphenyllactic acid in the serum is within 2–4 µM, and with sepsis, its concentration in the circulation increases and reaches the highest values (up to 10–100 µM or more); that is, it occurs in parallel with the imbalance of various processes in the body of a septic patient [34,35]. Thus, the results obtained shed light on the molecular mechanisms of the regulation of the inflammatory process involving the microbiota and its metabolites.

Thus, our research confirms the assumption that the regulation of a systemic inflammatory process is associated with the state of the microbiota and its metabolites. Our data show that the involvement of microbial metabolites in the regulation of the activity of COX is mainly due to their influence on the peroxidase activity of the enzyme. As a consequence, the redox state of cells, including the protection of thiol groups, allows them to obviate the deleterious effects of the inductors of oxidative stress and inflammation. Our results obtained under model conditions show a strong dependence of the direction of the action of microbial metabolites on the concentration of redox-active participants. The model system in which the intrinsic intermediate of the cyclooxygenase step, endoperoxide, was replaced by *t*-butyl-hydroperoxide made it possible to identify the factors that promote either the inhibition or activation of the enzyme.

## 4. Materials and Methods

### 4.1. Reagents and Chemicals

The colorimetric COX Activity Assay Kit (Cayman Chemical Company, Ann Arbor, MI, USA) was used in experiments. All other reagents were obtained from the Sigma–Aldrich Corporation (St. Louis, MO, USA).

### 4.2. Cell Cultures 

Human acute myeloid leukemia cell line THP-1 was obtained from the ATCC (Manassas, VA, USA). Cells were cultured in RPMI 1640 (Sigma-Aldrich, St. Louis, MO, USA) medium supplemented with 10% fetal bovine serum (Thermo Scientific, Waltham, MA, USA), 40 µg/mL gentamicin sulfate (Sigma-Aldrich, St. Louis, MO, USA), and 0.05 mM 2-mercaptoethanol (Sigma-Aldrich, St. Louis, MO, USA) at 37 °C in a humidified atmosphere of 5% CO_2_, as described previously [36,37]. 

Human peripheral blood CD14+ monocytes (HPBMs) were obtained from the CLS Cell Lines Service GmbH (Eppelheim, Deutschland). HPBMs were cultured in Human Blood Cell Culture Medium (CLS Cell Lines Service GmbH, Eppelheim, Deutschland) with 40 µg/mL of gentamicin sulfate (Sigma-Aldrich, St. Louis, MO, USA) at 37 °C in a humidified atmosphere of 5% CO_2_.

Monocyte-like THP-1ATRA cells were obtained via the treatment of THP-1 cells with ATRA. THP-1 cells were cultured in RPMI 1640 (Sigma-Aldrich, St. Louis, MO, USA) supplemented with 10% FBS (Thermo Scientific, Waltham, MA, USA) and 2 µM ATRA (Sigma-Aldrich, St. Louis, MO, USA) for 6 days. Then, cells were washed three times with RPMI 1640 and used for experiments.

The testing of cell cultures for mycoplasma infection was performed using the MycoFluor™ mycoplasma detection kit (Thermo Scientific, Waltham, MA, USA). The infection of cell cultures with mycoplasma was not detected.

### 4.3. Cell Lysate Preparation

Cells in cultures were activated by adding PMA (100 nM) one day before the experiment. Then, cells were washed with RPMI 1640 medium and used for experiments. Prior to obtaining cell lysates, the number of cells in culture in each experiment was determined using a TC20 Cell Counter (Bio-Rad, Hercules, CA, USA). This indicator was used for the normalization of the enzyme activity in lysates per million cells of the tested cell cultures. For cell lysate preparation, cells were collected via centrifugation (2000× *g* for 10 min at 4 °C). The cell pellet was homogenized in a cold buffer (0.1 M Tris-HCL, pH 7.8 with 1 mM EDTA) by a Dounce homogenizer (Sigma-Aldrich, St. Louis, MO, USA). Then, the cell lysate was centrifuged (10,000× *g* for 15 min at 4 °C) and the supernatant for the assay was stored on ice. Cell lysates were also obtained using the second method using ultra-sonication to disrupt cells. For this, the cells were subjected to the ultrasonic treatment at certain parameters (amplitude: 30; process time: 20 s; pulse on time: 5 s; pulse off time: 20 s) using a Q700 sonicator (Qsonica, Newtown, CT, USA). Destroyed cells were centrifuged (10,000× *g* for 15 min at 4 °C), and the supernatant was collected and stored on ice.

### 4.4. Determination of the Cyclooxygenase Activity and Influence of Microbiota-Related Metabolites on the Enzyme Activity in Cell Lysates

Cyclooxygenase activity was assayed using the colorimetric method by monitoring the oxidation of N,N,N’,N’-tetramethyl-p-phenylenediamine (TMPD) at 590 nm. Measurements were carried out in accordance with the instructions of the assay kit and the description of the method [38]. The influence of microbiota-related metabolites on COX activity was determined in a 96-well plate with the following order of additions of reaction components; the enzyme was incubated with the microbial metabolite for 5 min, followed by the addition of heme, arachidonic acid, and TMPD. The influence of microbiota-related metabolites on COX activity in cell lysates was determined in a similar mode: their incubation with the lysate preceded the addition of the substrate, the cofactor, and TMPD. Changes in the optical density of TMPD during the reaction were recorded using a microplate reader iMark (Bio-Rad, Hercules, CA, USA).

### 4.5. Statistical Analysis

The data given represent the means ± standard error of means (SEMs) from five experiments or are the typical traces of three to five identical experiments. Statistical significance was estimated via the Student’s *t*-test with *p* < 0.05 as the criterion of significance.

## Figures and Tables

**Figure 1 ijms-24-16244-f001:**
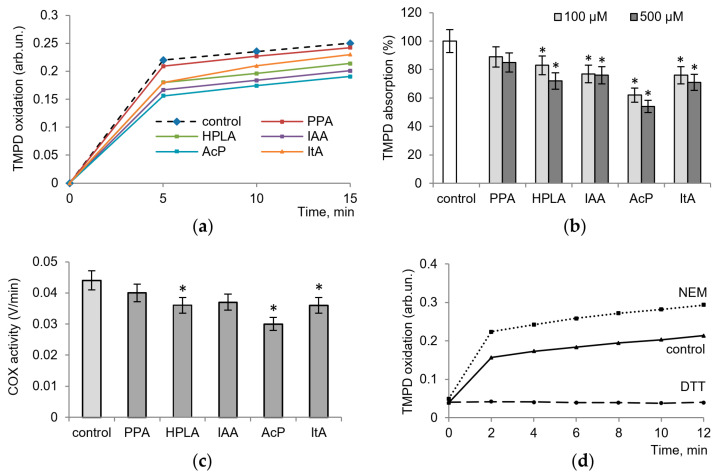
Influence of microbiota-associated metabolites and thiol reagents on commercially pure enzyme activity, as measured via TMPD oxidation. TMPD oxidation in the control and in the presence of 100 µM microbiota-associated metabolites (**a**), TMPD absorption in the control and in the presence of 100 µM and 500 µM microbiota-associated metabolites after 10 min of incubation (**b**), COX activity measured via the rate of TMPD oxidation per minute in the control and in the presence of 100 µM microbiota-associated metabolites (**c**), and the influence of NEM (25 µM) and DTT (250 µM) on TMPD oxidation (**d**). All measurements were performed in the presence of equal amounts of the commercially pure enzyme and arachidonic acid. An asterisk (*) indicates values that differ significantly from the control values (*p* < 0.05).

**Figure 2 ijms-24-16244-f002:**
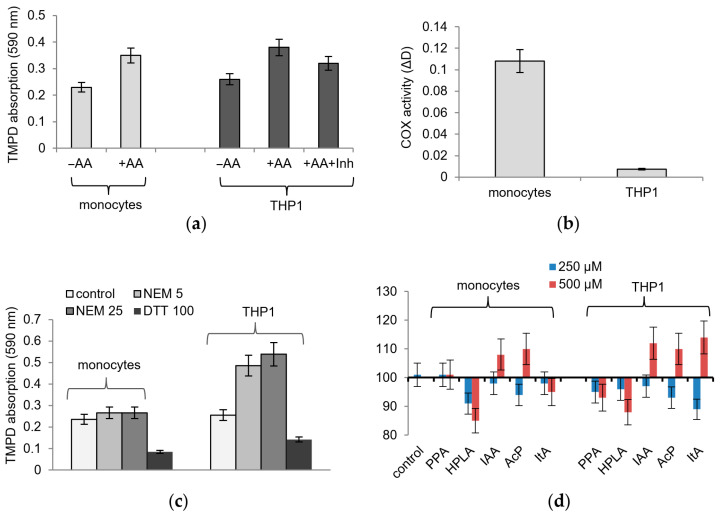
Influence of microbiota-associated metabolites and thiol reagents on COX activity in healthy and leukemic monocytes. An increase in TMPD absorption in response to the addition of arachidonic acid (AA) and the influence of the inhibitor SC-560 (**a**), COX activity as estimated via the difference in TMPD adsorption after 10 min of incubation of cell lysates with or without AA, per million cells (**b**), the influence of NEM and DTT on COX activity measured via TPMD absorption after 10 min of incubation with cell lysates (**c**), and the influence of microbiota-associated metabolites on COX activity (%) where the value of TMPD absorption after 10 min of incubation of cell lysates in the control is taken as 100% (**d**).

**Figure 3 ijms-24-16244-f003:**
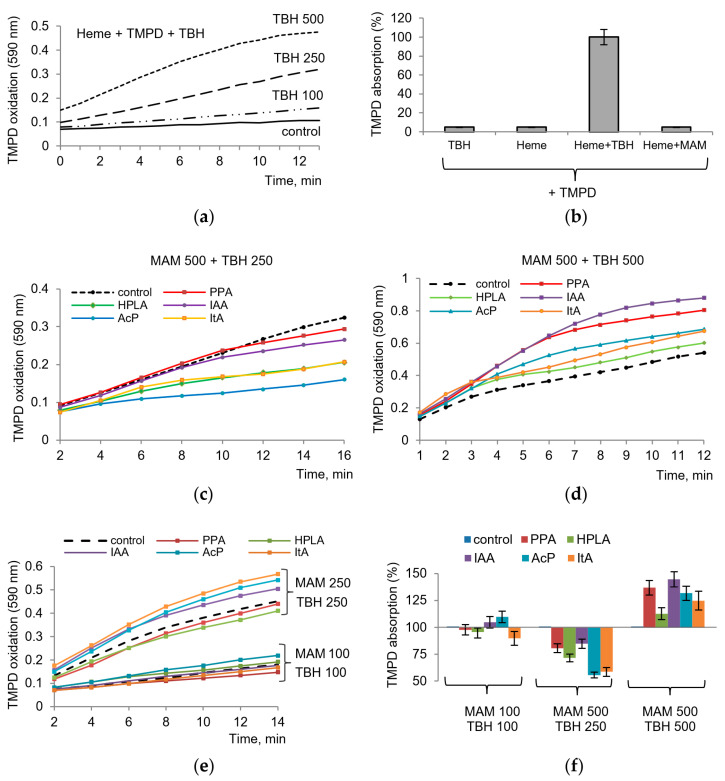
Influence of microbiota-associated metabolites on peroxidase activity in model conditions. An enhancement of TMPD oxidation with increasing TBH concentration (**a**), the role of each component in the induction of TMPD oxidation (**b**), TMPD oxidation at varying concentrations of TBH and microbiota-associated metabolites (MAM) (**c**–**e**), and the activation and inhibition of TMPD oxidation depending on the concentrations of TBH and microbiota-associated metabolites (MAM), where the absorption of TMPD after 10 min of incubation with TBH at each indicated concentration is taken as 100% (**f**).

## Data Availability

The datasets generated during and/or analyzed during the current study are available from the corresponding author on reasonable request.

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
