# Peer review of "Influence of Microbiota-Related Metabolites Associated with Inflammation and Sepsis on the Peroxidase Activity of Cyclooxygenase in Healthy Human Monocytes and Acute Monocytic Leukemia Cells"

_ijms, 2023, doi:10.3390/ijms242216244_

Round 1

Reviewer 1 Report

Comments and Suggestions for Authors

The article titled ‘Influence of microbiota-related metabolites associated with inflammation and sepsis on the peroxidase activity of cyclooxygenase in human healthy monocytes and acute monocytic leukemia cells’ studied the endogenous metabolites from human that potentially influence on the cyclooxygenase as a key to inflammation. The authors designed the study and tested a few human microbiota metabolites on their activity of inhibition COX on both pure enzyme and cell lysates.

I have some questions about the work:

1.       Line 132. The authors indicated that by observing the decreased activity of COX after exposure to certain metabolites is aligned with the trend after exposure to DTT while NEM active the activity, which indicates microbial metabolites through thiol pathways. In my opinion, it can’t be conclusive evidence of those metabolites influence the COX activity in that mechanism unless a thiol-blocking is added to the metabolite treatment and a thriving activity level is observed again.

2.       Figure 2d. data was all over the place and not sure if a conclusion can be made.

3.       Line 164, ‘As shown in Figure 3’…. Seems to be Figure 2C?

4.       Line 162, ‘As shown on Figure 2, both compounds had a strong opposite effect on the activity of the enzyme in cell lysates; namely, NEM activated and DTT inhibited TMPD oxidation (Figure 2d).’ Couldn’t locate the corresponding figure/results.

5.       Suggested to include all chemicals impurities as reference material.

Comments on the Quality of English Language

Minor revision is necessary

Author Response

We thank the reviewer for the useful remarks to our work. We have made appropriate corrections and additions to the manuscript.

  1. Line 132. The authors indicated that by observing the decreased activity of COX after exposure to certain metabolites is aligned with the trend after exposure to DTT while NEM active the activity, which indicates microbial metabolites through thiol pathways. In my opinion, it can’t be conclusive evidence of those metabolites influence the COX activity in that mechanism unless a thiol-blocking is added to the metabolite treatment and a thriving activity level is observed again.

In this context, we agree with you, but we do not conclude that the oxidation/reduction of thiol groups underlies the action of metabolites on COX activity. Our experiments in a model system showed a total dependence of the peroxidase reaction on the presence of the heme iron and the concentration of butyl-hydroperoxide. Microbial metabolites interfere with this reaction, i.e., act in the absence of any thiol groups. So, in our experiments, the addition of the blocker and the reductant of thiol groups reveals different redox states of these cells, which may contribute to the oxidation of TMPD as an electron donor. The specific role of thiol groups cannot be excluded, and we plan more detailed experiments along this line for the most important metabolites in further studies.

We have added a conclusion to the Discussion section in which we noted this aspect.

Conclusions. Thus, our research confirms the assumption that the regulation of a systemic inflammatory process is associated with the state of the microbiota and its metabolites. Our data show that the involvement of microbial metabolites in the regulation of the activity of COX is mainly due to their influence on the peroxidase activity of the enzyme. As a consequence, the redox state of cells, including the protection of thiol groups, allows them to obviate the deleterious effects of inductors of the oxidative stress and inflammation. Our results obtained under model conditions show a strong dependence of the direction of the action of microbial metabolites on the concentration of redox-active participants. The model system, in which the intrinsic intermediate of the cyclooxygenase step, endoperoxide, was replaced by t-butyl-hydroperoxide, makes it possible to identify the factors that promote either the inhibition or activation of the enzyme.

  1. Figure 2d. data was all over the place and not sure if a conclusion can be made.

From these data, we only conclude that microbial metabolites influence the reaction and determine which of them are the most active. Our next experiments under model conditions, where intrinsic endoperoxide is replaced by butyl-hydroperoxide, explain the observed variations by the dependence of the TMPD response on the concentration of peroxide and the concentration of the redox-active microbial metabolite

  1. Line 164, ‘As shown in Figure 3’…. Seems to be Figure 2C?

Yes, Fig. 2c, corrected

  1. Line 162, ‘As shown on Figure 2, both compounds had a strong opposite effect on the activity of the enzyme in cell lysates; namely, NEM activated and DTT inhibited TMPD oxidation (Figure 2d).’ Couldn’t locate the corresponding figure/results.

Fig. 2c, corrected

  1. Suggested to include all chemicals impurities as reference material.

All reagents used are from the kit; no other reagents were used. All microbial metabolites are also of the highest purity grade from Sigma. In addition, we have extensive experience in working with electron acceptors and carriers so that the necessary controls were made and presented in Fig. 3b. It can be seen that none of the components by itself causes the oxidation of TMPD. The same applies to microbial metabolites, each of which was tested separately.

We have added these sentences to the Results:

It can be seen that none of the components by itself causes the oxidation of TMPD. The same applies to microbial metabolites, each of which was tested separately.

Reviewer 2 Report

Comments and Suggestions for Authors

This is well written manuscript concerning huge challange nowadays - sepsis.

Concentrating on mechanisms underlying the pathology is required to improve the prevention and management of sepsis.

Some suggestions for the paper:

Introduction,

line 58-59: Pathological conditions cause microbiota disregulation and vice versa, also. Please, add information that this relationship is bidirectional, perhaps - gut dysbiosis may affect higher risk of chronic diseases.

line 49-50 - please deleta dot after "metabolism". The dot should be placed after the citation. Authors should carefully review the manuscript.

line 54-55, line 74-75 The abbreviations: IL-6, IL-1...  should be defined the first time they appear in main text. Authors should add abbreviations also IL-10, TNF-alpha and IFN-y.

line 100-101: Sentecne "The activity of COX..." is method rather and is not reqiured in the Introduction when authors explain the aim of the study.

Discussion:

In the first sentence of the discussion, Authors should remind the aim of the study for audience. 

In the paper, strengths and limitations of the study, as well as conclusions, should be added - these are highly important and key elements of research.

Statistical analysis:

line 365-266: In the study, there were more than two groups. Why authors apply t-test? I suggest apply the test for multiple comparision (ANOVA).

Author Response

We thank the reviewer for the useful remarks to our work. We have made appropriate corrections and additions to the manuscript.

Introduction,

line 58-59: Pathological conditions cause microbiota disregulation and vice versa, also. Please, add information that this relationship is bidirectional, perhaps - gut dysbiosis may affect higher risk of chronic diseases.

We added:

Besides, these relationships are bidirectional, since gut microbiota dysbiosis may provoke a higher risk of the onset of chronic diseases.

line 49-50 - please deleta dot after "metabolism". The dot should be placed after the citation. Authors should carefully review the manuscript.

We corrected the inaccuracy and carefully read the manuscript again.

line 54-55, line 74-75 The abbreviations: IL-6, IL-1...  should be defined the first time they appear in main text. Authors should add abbreviations also IL-10, TNF-alpha and IFN-y.

it was:

….stimulate the synthesis of the proinflammatory cytokines IL-6, IL-1, IL-27, and TNF-α

it became:

…….stimulate the synthesis of the proinflammatory cytokines, namely, Interleukins IL-6, IL-1, IL-27, and the tumor necrosis factor TNF-α.

it was:

….production of interleukin-10, the tumor necrosis factor-alpha and interferon-gamma

it became:

…..production of interleukin-10 (IL-10), the tumor necrosis factor-alpha (TNF-α), and interferon-gamma (IFN-y)

line 100-101: Sentecne "The activity of COX..." is method rather and is not reqiured in the Introduction when authors explain the aim of the study.

Now:

Their influence on COX activity was tested on the commercial pure enzyme, cultured healthy monocytes, and the human acute monocytic leukemia cell line THP1.

Discussion:

In the first sentence of the discussion, Authors should remind the aim of the study for audience.

We have added the following text to the beginning of the Discussion:

The development of a systemic inflammatory reaction in the human body can be observed in various nosological forms of diseases, including local infections of the skin and soft tissues, pneumonia, osteomyelitis, and others, as well as tissue damage (trauma, surgery). An increase in clinical and laboratory signs of inflammation is always regarded as a progression of the pathological process, since it is associated with the development of certain complications; and in the most severe cases, it is accompanied by the development of multiple organ dysfunctions and sepsis with the risk of an unfavorable outcome. Given extensive evidence on the role of the structural components of the bacterial cell wall (LPS, etc.), as well as immunological, genetic and many other etio-pathogenetic factors involved in triggering systemic inflammation and sepsis, today very little is known about the role of  the products of microbial metabolism  in the regulation of the inflammatory process. Our results showed the participation of microbiota-related metabolites in the regulation of COX activity as a key enzyme of inflammation.

In the paper, strengths and limitations of the study, as well as conclusions, should be added - these are highly important and key elements of research.

Conclusions. Thus, our research confirms the assumption that the regulation of a systemic inflammatory process is associated with the state of the microbiota and its metabolites. Our data show that the involvement of microbial metabolites in the regulation of the activity of COX is mainly due to their influence on the peroxidase activity of the enzyme. As a consequence, the redox state of cells, including the protection of thiol groups, allows them to obviate the deleterious effects of inductors of the oxidative stress and inflammation. Our results obtained under model conditions show a strong dependence of the direction of the action of microbial metabolites on the concentration of redox-active participants. The model system, in which the intrinsic intermediate of the cyclooxygenase step, endoperoxide, was replaced by t-butyl-hydroperoxide, makes it possible to identify the factors that promote either the inhibition or activation of the enzyme.

line 365-266: In the study, there were more than two groups. Why authors apply t-test? I suggest apply the test for multiple comparision (ANOVA).

We compared only two groups of data, namely, on the control activity of the enzyme and its activity in the presence of microbial metabolites, each of which being tested separately.

Round 2

Reviewer 1 Report

Comments and Suggestions for Authors

The authors addressed all the comments, the manuscript now is qualified to be published in IJMS.

Comments on the Quality of English Language

Minor changes is needed.

Author Response

We have carefully looked through the English and identified and corrected some errors and inaccuracies.

Thank you very much for your attention and comments, they were very useful.